# Enteric Pathogen Diversity in Infant Foods in Low-Income Neighborhoods of Kisumu, Kenya

**DOI:** 10.3390/ijerph16030506

**Published:** 2019-02-12

**Authors:** Kevin Tsai, Sheillah Simiyu, Jane Mumma, Rose Evalyne Aseyo, Oliver Cumming, Robert Dreibelbis, Kelly K. Baker

**Affiliations:** 1Department of Occupational and Environmental Health, University of Iowa, Iowa City, IA 52246, USA; meng-hsien-tsai@uiowa.edu; 2Center of Research, Great Lakes University of Kisumu, Kisumu 40100, Kenya; sheillahshie@gmail.com (S.S.); jnmumma@gmail.com (J.M.); evalyneaseyo6@gmail.com (R.E.A.); 3Department of Disease Control, London School of Hygiene and Tropical Medicine, London WC1E 7HT, UK; oliver.cumming@lshtm.ac.uk (O.C.); robert.dreibelbis@lshtm.ac.uk (R.D.)

**Keywords:** food, sanitation, infants, milk, pathogen presence, pathogen diversity, TaqMan Array Card

## Abstract

Pediatric diarrheal disease remains the second most common cause of preventable illness and death among children under the age of five, especially in low and middle-income countries (LMICs). However, there is limited information regarding the role of food in pathogen transmission in LMICs. For this study, we examined the frequency of enteric pathogen occurrence and co-occurrence in 127 infant weaning foods in Kisumu, Kenya, using a multi-pathogen PCR diagnostic tool, and assessed household food hygiene risk factors for contamination. Bacterial, viral, and protozoan enteric pathogen DNA and RNA were detected in 62% of the infant weaning food samples collected, with 37% of foods containing more than one pathogen type. Multivariable generalized linear mixed model analysis indicated type of infant food best explained the presence and diversity of enteric pathogens in infant food, while most household food hygiene risk factors considered in this study were not significantly associated with pathogen contamination. Specifically, cow’s milk was significantly more likely to contain a pathogen (adjusted risk ratio = 14.4; 95% confidence interval (CI) 1.78–116.1) and more likely to have higher number of enteric pathogen species (adjusted risk ratio = 2.35; 95% CI 1.67–3.29) than porridge. Our study demonstrates that infants in this low-income urban setting are frequently exposed to diarrhoeagenic pathogens in food and suggests that interventions are needed to prevent foodborne transmission of pathogens to infants.

## 1. Introduction

Even though the incidence of pediatric diarrheal diseases is declining worldwide, they remain the second most common cause of preventable illness and death among children under the age of five [1], responsible for approximately 800 million illnesses and 800,000 deaths in 2010. Approximately 90% of this disease burden is concentrated in children under the age of five in low- and middle-income countries (LMICs) [2,3]. Diarrheal infections are caused by a diverse range of enteric pathogens that infect children as early as birth [4]. Children infected with enteric pathogens can potentially suffer long-term adverse effects to their physical and cognitive development and future socio-economic status [5,6].

There is increasing recognition that consumption of pathogen-contaminated food is an important exposure pathway for diarrheal disease in children in LMICs [7,8]. Contaminated food causes an estimated 582 million cases of illness, death, or disability-adjusted life years each year globally, with young children and Africans bearing most of the foodborne disease burden [2]. The risk of diarrheal disease typically increases as infants transition from exclusive breastfeeding to consumption of weaning foods and water due to both decreases in passive protection from maternal breastmilk and more exposure to contaminated food [9,10]. Many caregivers worldwide struggle to exclusively breastfeed up to six months of age, resulting in infants being provided weaning food instead of breast milk before 6 months of age [11,12,13]. Thus, a premature transition from exclusive breast feeding to weaning foods may be especially important as one of the earliest causes of enteric infection [14]. Little is known about how often infants in LMICs are exposed to pathogens via food, and which risk factors should be targeted to reduce food-related exposure of children to enteric pathogens. In addition, infants’ diets become more diversified as they develop, and each of these additional food types may pose different exposure risks for different enteric pathogens [15]. More evidence is needed to understand which risk factors should be targeted to reduce food-related exposure of children to enteric pathogens in LMIC settings [16].

While foodborne transmission of enteric pathogens into the food supply chain is rigorously monitored in high-income countries (HICs) via regulatory authorities [17], food safety is frequently not monitored and regulated well in LMICs [7]. Many common infant weaning foods, like cow’s milk, are sourced from outside the household. Unsanitary and unregulated farm and market practices can result in contamination of milk by human or animal feces, well before entry to the household [18]. Reliance on unsanitary water to prepare weaning foods is common in LMICs [10]. In addition to sub-optimal water and market food supplies, insufficient hand washing and sterilization of food preparation areas, improper cooking temperature of infant food, storage of perishable foods at ambient temperatures, and storage of food in containers open to flies [7,19,20] can introduce additional microbial contamination in the household.

This study aimed (1) to describe the frequency and diversity in enteric pathogen contamination of infant weaning foods in low-income, neighborhoods of Kisumu, Kenya, and (2) to identity the leading environmental conditions and behaviors that contribute to pathogen presence and absence and higher pathogen diversity. The methodologies described and applied in this paper are useful for future research on foodborne illnesses in LMICs. Furthermore, our findings inform public health and healthcare professionals as a basis for prevention of pediatric diarrheal diseases in LMICs.

## 2. Materials and Methods

### 2.1. Study Setting/Ethical Consideration

This exposure assessment study was conducted as part of formative research aimed at developing and testing an infant hygiene intervention to inform the development and evaluation of an infant weaning food hygiene intervention in Kisumu, Kenya. Kisumu is a city in the western region of Kenya, with a projected population of approximately 490,000 people by 2017 (Kisumu County Integrated Development Plan 2013–2017). The study site includes four villages of a low-income peri-urban neighborhood in Kisumu. This infant weaning food hygiene intervention will be evaluated as part of the Safe Start study, a cluster-randomized controlled trial (Clinical Trials identifier: NCT03468114) involving Great Lakes University of Kisumu, Kenya (GLUK), the London School of Hygiene and Tropical Medicine (LSHTM), and the University of Iowa (UI). The study was approved by the scientific and ethical review committees at the GLUK (Ref. No. GREC/010/248/2016), LSHTM (Ref. No. 14695), and UI (IRB ID 201804204).

### 2.2. Study Design

Eight community health volunteers (CHVs) who served the four neighborhoods in our study area facilitated the recruitment of participants. First, CHVs conducted a household census with the research team in December 2016 to generate a list of all infants less than nine months of age that were living in each CHV’s catchment area. Then, the list of households was randomly sorted, and in January CHVs and enumerators approached each house to verify infant eligibility, obtain consent to participate in the study, and perform data collection and food sampling. CHVs maintained surveillance of their respective catchment area through May 2017 to identify new infants as they became age-eligible, and to approach their caregivers about participation in the study.

Eligibility of a household was defined as having an infant between three and nine months of age, verified by reviewing the infant’s birth identity card, who was being fed supplemental food in addition to or in replacement of breastfeeding. Exclusion criteria included refusal to participate, inability to produce infant health card for verification of age, or caregiver reporting that the infant was exclusively breastfed and did not eat other food or liquid. Upon verification of eligibility and availability of food for sampling, the child’s primary caregiver provided consent to participate in the study in the presence of the CHVs. The study was described in the caregiver’s natural language, and a signed copy of the consent form was left for her records.

### 2.3. Data and Sample Collection

After agreeing to participate in the study, caregivers were interviewed about household status, their level of education; access to water, sanitation, and hygiene resources; and key infant weaning food preparation, storage, and feeding practices. Caregivers were then asked to provide approximately five grams of already-prepared infant food of any type fed to the child that day. The timing of food preparation for infants varied throughout the day, so the field team scheduled follow-up visits with households at times when food would be available. Food was placed into a sterile, labeled WhirlPak bag (Sigma-Aldrich, St. Louis, MO, USA) by the caregivers, using whatever means (fingers, utensil) that the caregiver normally used for handling the child’s food. Food was placed on ice packs in a cooler and was transported to the laboratory for processing within six hours of collection.

### 2.4. Nucleic Acid Extraction

All food samples were processed by following the manufacturer's instructions for the ZymoBIOMICS™ DNA/RNA extraction mini-kit (Zymo Research, Irvine, CA, USA) for DNA and RNA parallel purification. A 250 mg food sample was measured into the Zymo-Shield tube, vortexed until blended, and stored at 4 °C. Samples were then transported in a cooler at ambient temperature to the University of Iowa for the remainder of the extraction. A subset of samples (*n* = 77) was spiked with 5 µL of 1.8 × 10^9^ colony-forming units (CFU)/mL of live bacteriophage MS2 to serve as an extrinsic process control to assess for RNA degradation as a function of storage and transport conditions. Once purified DNA/RNA was obtained, it was stored at −80 °C until further analysis.

### 2.5. Inhibition Screening/Preamplification

DNA and RNA extracts from the samples (6 µL each) were screened for evidence of inhibition with the QuantiFast Pathogen PCR +IC Kit and QuantiFast Pathogen qRT-PCR+ IC kit (Qiagen, Hilden, Germany) on a QuantStudio real-time PCR system (Thermo Fisher, Waltham, MA, USA). Seventy-seven samples were screened for inhibition, defined as having amplification of the RNA internal control over cycle threshold value (CT value) of 34 in a sample according to the manufacturer’s protocol. The QuantiFast Pathogen qRT-PCR +IC Kit (Qiagen, Hilden, Germany) was used for pre-amplification PCR. For each sample, a total volume of 12 µL of DNA and RNA extract (6 µL each) was mixed with a master mix containing 5 µL of 5× Quantifast Pathogen MM, 0.25 µL of 100× Quantifast Pathogen RT Mix, 0.5 µL of 50× high ROX dye solution, 0.15 µL of ultrapure 50 mg/mL bovine serum albumin (BSA), 2.5 µL of 0.2× custom TaqMan pre-amplification primer and probe pool (Appendix 1), 2.5 µL of internal control assay, and 2.0 µL of internal control RNA. If extracts were determined to be inhibited during the inhibition screening, the inhibited extracts would undergo 1:10 dilution before mixing with the pre-amplification master-mix. The cycling conditions for the pre-amplification PCR were: holding stage of 50 °C for 20 min and 95 °C for 5 min, followed by 44 cycles of 95 °C for 15 s and 60 °C for 30 s [21]. Preamplification PCR was completed through an Eppendorf Thermocycler (Eppendorf, Hamburg, Germany). All the completed pre-amplified samples would undergo a 1:10 dilution with nucleic acid-free water before proceeding to TaqMan quantitative PCR card analysis.

### 2.6. TaqMan Array Card Analysis

Primers and probes for a total number of 37 gene targets of pathogen of interest in the TaqMan assays are listed in Appendix A. The Ag-Path-ID One-Step RT-PCR kit (Thermo Fisher, Waltham, MA, USA) was used for the TaqMan card analysis. For each sample, 40 µL of re-amplified DNA/RNA extract (in 1 to 10 dilutions with nucleic acid-free water) were mixed with 50 µL of 2× RT-buffer, 4 µL of 25× AgPath enzyme, and 6 µL of nucleic acid-free water. All the TaqMan runs were completed in a ViiA7 instrument (Thermo Fisher, Waltham, MA, USA), and the cycling conditions were: 45 °C for 20 min and 95 °C for 10 min, followed by 45 cycles of 95 °C for 15 s and 60 °C for 1 min. Amplification of a pathogen-specific gene target was used to define a sample as positive for the presence of that pathogen. If multiple gene targets were used to detect different one type of pathogen (norovirus, Enteroaggregative *E. coli* (EAEC), Enteropathogenic *E. coli* (EPEC), Enterotoxigenic *E. coli* (ETEC), Shiga toxin-producing *E. coli* (STEC) amplification of either gene resulted in a sample being considered positive. Two virulence gene indicators were used to detect pathogenic bacteria on the card, so in this manuscript samples were considered positive for the overall species of bacterial pathogen if either gene was detected.

### 2.7. Data Analysis

There were two primary outcomes assessed during analysis. First, a binary indicator was defined based on the presence of one or more target pathogens detected in the sample (any-path). Second, pathogen diversity was calculated by summing the total number of target pathogens types detected in the sample (sum-path). Caregiver education level and sampling month were selected a priori as potential confounders of infant health and caregiver food preparation practices [22] and included in all analyses. Proposed risk factors for food contamination by enteric pathogens were selected based upon general household conditions that could lead to the introduction (e.g., animals near food, feces, flies) or sustained presence (e.g., floor type) of feces with pathogens in the food preparation and feeding area, and food-specific practices (e.g., storage container, handwashing station presence and usage) that could influence whether pathogens in the preparation and feeding areas could be introduced into food (Figure 1). We documented ownership of domestic animals (yes/no) and whether animals were kept inside the household (yes/no). In food preparation and feeding areas specifically, we recorded permeable (dirt, grass) versus non-permeable (concrete, linoleum, metal) floor type, the presence of flies (yes/no), feces (yes/no), and a handwash station (yes/no). Food hygiene practices included sharing eating containers with family (yes/no) and the type of food container (bottle/jug, covered container, uncovered container, fresh food, and thermos). Food type was not specified ahead of time and was based upon whatever caregivers were feeding the child, which ultimately included cow’s milk, porridge (grains, water, and sometimes with milk), and “other” less common foods (tea, mashed potato, bread, beans). Due to infrequent (<5%) detection rates for tea, water, and other foods, these types were combined into one “other” group for the single and multivariable analysis to ensure model convergence.

Analyses were completed through SAS software version 9.4 (SAS Institute, Cary, NC, USA). Separate generalized linear mixed models (GLMMs) were developed for the binary indicator of any pathogen detection (any-path model) and the pathogen diversity measures (sum-path models) to assess relationships between environmental and behavioral risk factors and primary outcomes. For the any-path model, the log link and binomial distribution specifications were used, and regression results were converted to risk ratios. For the Sum-path models, Poisson, zero-inflated Poisson, negative binomial, and zero-inflated negative binomial distributions were evaluated, and the log link and negative binomial distribution family was ultimately determined to best fit the distribution of outcome data. Regression results were converted to risk ratios.

Both sets of models followed the same two-stage process. Including all variables in the model was deemed statistically implausible due to a low events per variable ratio and convergence problems caused by sparse data for categories of some variables of interest [23]. Thus, the model selection process for this study was based on the Hosmer and Lemeshow method for iterative and purposeful selection of covariates [24]. First, bivariate associations between environmental and behavioral risk factors were determined. Risk factors with p-values smaller than 0.30 in the bivariate testing were included in the multivariable analysis to achieve a balance between including important adjustment variables in the model versus obtaining numerically stable estimates and standard errors. We then followed a backwards selection process. The Akaike information criterion (AIC) score was noted for the model with all selected-in variables [25]. Then variables were removed individually, and the AIC score was recorded. Variables retained in the final models were selected based upon the model with the lowest AIC score, adjusted for educational level of the caregiver and month of sampling.

## 3. Results

### 3.1. Demographics of Caregivers/Infants and Household Hygiene Conditions

One hundred and twenty-seven households (caregivers/infant dyads) participated in this study. Seventy-seven households were enrolled in January following the initial census and recruitment of all children between 3 and 9 months of age, and another 30 and 20 households were enrolled in March and May, respectively (Table 1). The study population was comprised of 45% male and 54% female infants born between March 2016 and December 2016. Most infants of caregivers were over 6 months old (76%). Among the caregivers who provided the study samples, 83% were married. Half of the caregivers (50%) had only a primary education, whereas 21% had some secondary education and 30% completed secondary education. Almost half (47%) of the caregivers who reported their employment status were unemployed (Table 1). There was variability in number of households enrolled across the four CHV catchment areas due to differences in number of eligible infants for recruitment, inability to locate the caregivers after census, refusals to participate, ability to verify infant’s age, or refusal to provide food samples after consent and participation in the survey.

The most common infant food types were porridge and cow’s milk, followed by tea, and “other” food (example: flour bread, mashed potatoes, or beans) (Table 2). Food type did not vary for infants <6 months of age versus those older than 6 months (chi-squared, *p* = 0.12) when food types were categorized as milk, porridge, and non-milk/porridge (tea, water, flour bread, mashed potatoes, or beans were grouped due to low frequencies per category). Most of the households did not have a handwashing station in their food preparation and/or feeding area. Flies were observed in one third of household food preparation and feeding areas, and animal feces were observed in 8% of household food preparation and feeding areas. Non-permeable floors in food preparation and feeding areas were most common.

### 3.2. Pathogen Distribution and Diversity in Infant Weaning Foods

Assessment of the quality of DNA and RNA extracted from infant food is reported in Appendix A. DNA and RNA of 13 different types of bacterial, viral, and protozoa enteric pathogens was detected in 79 of the 127 (62%) infant weaning food samples collected over the three-month span (Table 3). The most commonly detected pathogens were *Aeromonas hydrophila* (20%), Enterohemorrhagic *E. coli* 0157 *(EHEC)* (17%), Enteropathogenic *E. coli (EPEC)* (17%), Enterotoxigenic *E. coli (ETEC)* (13%), *Adenovirus 40/41* (12%), and non-*parvum/*non-*hominus Cryptosporidium spp*. (10%), with eight other pathogens occurring in less than 10% of overall infant weaning food samples (Table 4). Infant food samples collected during March have higher raw contamination rate than food collected during January and May (90%, 52%, and 60%, respectively). ETEC and non-*parvum* or *hominus Cryptosporidium spp* were detected frequently in January. In March, detection frequencies for Adenovirus 40/41, EPEC, EHEC 0157, STEC, EIEC/*Shigella spp*., and *C. difficile* were highest. *A. hydrophila* were detected frequently in May. A median of one pathogen per sample (standard deviation of 1.58; range of 0 to 9 pathogen) was detected, with 37% of foods being co-contaminated by more than one pathogen type.

### 3.3. Risk Factors for Enteric Pathogen Presence in Infant Weaning Food

Food type, the infant sharing eating containers with other family members, and feces in preparation area were associated at *p* < 0.3 with presence of any pathogen in the bivariate analysis, and were included in the multivariable analysis (Table 4). Sharing an eating container did not improve model fit and was removed. In the final multivariable model, cow milk was significantly more likely to contain an enteric pathogen when compared with porridge, but non-milk/porridge foods were not statistically different from porridge. Pathogens were detected twice as often in milk (95%, *n* = 19/20) as porridge (56%, *n* = 45/81) and non-milk/porridge foods (56%, *n* = 15/26). Observation of feces in preparation area was statistically associated with a lower risk of pathogen presence compared to feces not observed. 

Food type, handwashing station in preparation area, sharing eating containers with family members, and owning animals were associated with higher pathogen diversity at *p* < 0.3 in bivariate analysis and were considered in the multivariable model (Table 5). Food type was the only variable retained in the final model for explaining pathogen diversity. Pathogen diversity was 2.35 times higher in milk than in porridge, whereas non-milk/porridge foods trended towards lower levels of diversity.

## 4. Discussion

Estimates of the importance of food as an enteric infection pathway for young children in LMICs are limited by the absence of primary data on food outbreaks and frequency of food contamination by enteric pathogens, especially with respect to weaning foods provided to infants [17]. This study demonstrated that infants as young as three months of age in informal settlements of Kisumu ingest food contaminated by a variety of different types of enteric pathogens. Our qPCR-based enteric pathogen detection frequency of 62% is similar to what has been reported for frequency of fecal indicator bacteria in infant food in similar high disease burden settings, such as Bangladesh (40–58%), Indonesia (45%), South Africa (70%), India (56%), and Peru (48%) [2,26,27,28,29,30]. We expand upon these studies to show that a substantial number of infants ingest food contaminated by multiple types of enteric pathogens. 

Studies vary in their conclusions as to which pathogens cause the most foodborne enteric disease in LMICs, e.g., Norovirus, *Campylobacter spp*., *S. enterica*, ETEC, EPEC, *Giardia lamblia*, and *Shigella spp*. are all attributed with a substantial amount of foodborne illness or death [2,31]. The etiology of foodborne disease may vary year to year, or month to month as suggested by our study, which has implications for ranking the priority foodborne pathogens in settings where outbreak or food monitoring information is limited. Many types of pathogens were detected in food during our 5-month study in Kisumu, with *A. hydrophila* being the most common pathogen, followed by EHEC O157, ETEC and EPEC, and human adenovirus 40/41. Aeromonas is extremely common in the environment, including foods, but is not considered a priority foodborne pathogen [32]. However, EHEC O157 is notorious as a deadly cause of foodborne epidemics, and the emergence of so many O157-positive food samples in March alone suggests potential for a foodborne outbreak. March is the onset of the rainy season in Kisumu. The increased detection of multiple pathogens during this month may reflect an influence of seasonality on foodborne transmission risks in Kenya. This foodborne danger would have been missed had we sampled in a narrower timeframe.

We demonstrated that the risk of pathogen exposure for an infant can vary by type of weaning food, which has important implications for designing interventions. Cow’s milk was significantly more likely to be contaminated by one or multiple types of enteric pathogens compared to other common infant foods, such as porridge. For many in urban and rural Kenya, raw milk is more affordable and accessible than pasteurized milk [33]. However, raw milk can be easily contaminated during production at farms by animal urine and feces, dirt, flies, adulteration with untreated water, and improperly cleaned containers [34]. In addition, the packaging, storage, distribution, and marketing of milk are not rigorously regulated and monitored in Kenya, leading to additional points where unhygienic conditions can introduce contamination [35]. Urban populations often encounter milk adulterated by water [36,37]. Therefore, pasteurization of milk at the point of sale or during food preparation in the household may be critical for rendering milk safe to drink. 

After sale, household food preparation, feeding, and storage conditions can contribute to new sources of infant food contamination [19]. In Kenya, milk is often consumed in liquid form, as well as is added to a variety of infant foods. Depending upon how the milk is provided to the infant, it may or may not receive proper treatment to eliminate microbial pathogens. If caregivers perceive milk to be safe due to prior pasteurization, they may not treat it further. In this study, we classified food by its primary ingredient, however tea and porridge (made from maize meal, sorgum meal alone, or sorgum mixed with millet meal) are also typically made with milk. If milk is added to infant foods, it may be reheated as a part of the cooking process or can be added to food after the preparation process without reheating. If the latter is more common, some of the pathogens detected in tea and porridge in this study may have come from milk sources. Milk is an optimal growth medium for bacteria, and may be particularly sensitive to cross-contamination from unclean surface, hands, and flies or uncovered and unclean containers. Public health interventions targeting safety of milk products may be particularly effective for reducing foodborne diarrheal diseases in infants in LMICs.

Household water, sanitation, and hygiene (WASH) interventions have been suggested as key in combating enteric pathogen transmission and infections [16,38,39]. While we found some domestic food hygiene risk factors were associated with enteric pathogen presence and diversity in bivariate analysis, most of these associations reduced in magnitude and did not improve model fit after adjustment for food type. Counterintuitively, observation of feces in the food preparation area—a rare situation to begin with—was associated with a lower likelihood of pathogen presence after model selection, rather than higher child food contamination as expected [19]. This association may be caused by unmeasured confounding factors or reactivity of some caregivers who were aware of the purpose of our visit. Timing household visits to coincide with food availability is logistically challenging unless caregivers store food for infants for prolonged periods of time. Thus, if food was not present during our first visit, we had to work carefully with households to time our follow-up visits to coincide with when they would have food for the child. Some caregivers may have reacted to the presence of feces in their preparation area before our visit and contaminated the infant’s food in the process of removing it. The inability to determine causality is a limitation of our cross-sectional design. The lack of association between other well-known risk factors of bacterial contamination in infant food could be caused by lack of statistical power to detect smaller effect sizes, although does not detract from the dominant role of food type in explaining pathogen detection. Analysis with a larger sample size are underway to improve knowledge about foodborne pathogen transmission in Kisumu. 

One of the strengths of our study is that we examined food for pathogens, rather than bacterial indicators, using rigorous microbiological protocols to ensure quality of data was preserved from field labs in Kisumu to molecular labs in Iowa. Fecal indicators, which are typically used as a proxy for determining risk from fecal pathogens, are nonspecific and often do not correlate well with viral, bacterial, and protozoan pathogens [40]. Addressing the need for information on infant food contamination in LMICs required finding an effective microbial testing method that enabled quantitative and target-specific measuring of a broad array of the most common types of diarrhea pathogens in infant food. Even though qPCR is frequently applied for the quantitative detection of pathogen presence in foodborne outbreak analysis [41,42,43], it has not been widely applied in food samples in LMICs. Our methods are novel in their ability to detect a wide array of pathogens simultaneously. The qPCR approach is also a limitation, since qPCR-determined concentrations may detect non-infectious organisms that cannot cause disease. We are not certain what fraction of the PCR-detected pathogens in this study were viable, viable-but-non-culturable, or dead microbial organisms. However, the distinct variability in contamination patterns in infant food and the consistency of cow’s milk as a risk factor for pathogen presence suggests qPCR was a valid approach for identifying infant food risk factors.

Several challenges had to be overcome for measuring and analyzing infant weaning food contamination in this study, challenges which apply to many LMIC settings. First, multi-target enteric pathogen detection capability is limited in LMICs due to limited laboratory facilities, requiring samples to be sent to specialized labs for precise analysis. We minimized sample degradation risks by using a ZymoBIOMICS™ DNA/RNA extraction kit that allowed us to preserve samples at ambient temperature for storage, transport, and extraction [44]. This makes the method more desirable for use in laboratory-limited LMICs or any field-based scenario, as samples can be shipped to an equipped laboratory for processing with ease. The high rates of recovery of MS2 virus spiked into samples before storage and transport confirmed that we experienced no loss in nucleic acid using this process. Second, the wide variety of physical and chemical properties of different food types makes optimization of microbial food testing protocols complex, especially if the goal is to measure multiple types of pathogens (Appendix A) [45]. In addition, the presence of inhibitors can impact qPCR performance [46]. We pre-piloted all protocols to confirm that protocols for DNA and RNA recovery of spiked pathogens was not affected by food type, then rigorously evaluated each sample for signs of inhibition prior to qPCR analysis. Low inhibition rates and low variability in MS2 Ct values across all food types showed that the Zymo extraction kit can produce high-quality nucleic acids free of inhibition from foods (Appendix A) [47,48,49]. Third, pathogens may be present in food at concentrations that are lower than the methodological limit of detection, which results in misclassification of some pathogen-positive samples as uncontaminated. Pre-amplification increased the concentration of starting content before conducting the quantitative measurement step of PCR.

## 5. Conclusions

Foodborne disease transmission of enteric pathogens may contribute substantially to the global diarrheal disease burden, yet receives limited attention. Our evidence highlights a need for more interventions targeting safe preparation and storage of infant foods, particularly high-risk foods such as milk. The ongoing Safe Start study in Kisumu is evaluating whether behavior improvements in caregiver food preparation, feeding, and storage behaviors can reduce enterococcus contamination in infant food and enteric pathogen infections in infants during weaning. Alongside interventions aiming to improve food hygiene practices of caregivers, interventions targeting hygienic milk handling and storage at the point of sale and among manufacturers may be needed to address upstream risks. The intersecting Market to Mouth study will contribute more information about the role of locally sold milk sources on pathogen contamination of infant food and the ability of the Safe Start intervention to mitigate enteric pathogen contamination passed via the food system.

## Figures and Tables

**Figure 1 ijerph-16-00506-f001:**
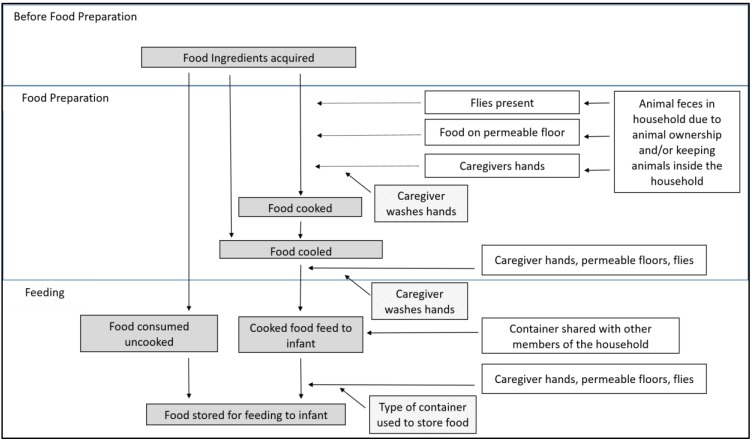
Conceptual model of hazard points (white boxes, right) where enteric pathogen contamination could enter the food preparation and feeding process for infants (grey boxes, left), and the mitigating actions that could prevent contamination transmission (gridded boxes, center).

**Table 1 ijerph-16-00506-t001:** Socio-economic demographic statistics for 127 caregivers and infant dyads in Kisumu.

Variable	Category	Number of Samples	Percentage
Infant gender	Male	58	46
	Female	69	54
Marriage status of caregiver	Married	108	85
	Single	17	13
	Divorced	2	2
Education level of caregiver	Some primary	27	21
	Complete primary	35	28
	Some secondary	27	21
	Complete secondary	38	30
Occupation	Agriculture	1	1
	Domestic service	8	6
	Not employed	60	47
	Managerial	9	7
	Sales and service	33	26
	Other	6	4
	Missing	10	8
Village	A	34	27
	B	35	28
	C	24	19
	D	34	27
Infant age	3–6 months	30	24
	More than 6 months	97	76

**Table 2 ijerph-16-00506-t002:** Demographic statistics for 127 infant foods in households in Kisumu by household and food hygiene conditions.

Variable	Categories	Number of Samples	Percentage
Food type	Milk	20	16
	Porridge	81	64
	Non-milk or porridge combined	26	20
	Tea	7	6
	Water	13	10
	Other *	6	5
Container type	Bottle/feeding bottle/jug	53	42
	covered	26	20
	Fresh food	13	10
	Thermos	24	19
	Uncovered	11	9
Month of sampling	January	77	61
	March	30	24
	May	20	16
Owning animals	Yes	43	34
	No	84	66
Keeping animals inside	Yes	78	61
	No	39	31
	Missing data	10	8
Sharing eating containers with family members	Yes	43	34
	No	84	66
Food preparation area			
Floor type in preparation area	Permeable floor	26	20
	Non-permeable floor	101	80
Flies in preparation area	Yes	40	32
	No	77	61
	Missing data	10	8
Animal feces in preparation area	Yes	10	8
	No	117	92
Handwashing station in preparation area	Yes	26	20
	No	101	80
Feeding area			
Floor type in feeding area	Permeable floor	22	17
	Non-permeable floor	105	83
Flies present in feeding area	Yes	40	31
	No	77	61
	Missing data	10	8
Animal feces present in feeding area	Yes	10	8
	No	117	92
Handwashing station in feeding area	Yes	19	15
	No	108	85

* “Other” includes tea, bread, mashed potatoes, and beans.

**Table 3 ijerph-16-00506-t003:** Type of detected pathogens in all infant foods overall, and by month.

Any Type of Pathogen	Overall (*n* = 127)	January (*n* = 77)	March (*n* = 30)	May (*n* = 20)
	*n* (%) Positive	*n* (%) Positive	*n* (%) Positive	*n* (%) Positive
	79 (62)	40 (52)	27 (90)	12 (60)
Virus				
Adenovirus 40/41	15 (12)	3 (3)	10 (33)	2 (10)
Adenovirus Hexon	6 (5)	1 (1)	3 (10)	2 (10)
Norovirus	9 (7)	4 (5)	3 (10)	2 (10)
Sapovirus	1 (1)	1 (1)	0 (0)	0 (0)
Bacteria				
EAEC	6 (5)	4 (5)	2 (7)	0 (0)
EPEC	21 (17)	3 (4)	15 (50)	3 (15)
ETEC	17 (13)	13 (17)	3 (10)	1 (5)
EHEC O157	21 (17)	0 (0)	21 (70)	0 (0)
STEC	5 (4)	0 (0)	5 (17)	0 (0)
EIEC/Shigella	7 (6)	4 (5)	3 (10)	0 (0)
*A. hydrophila*	25 (20)	12 (16)	5 (17)	8 (40)
*B. Fragilis*	1 (1)	0 (0)	0 (0)	1 (5)
*C. difficile*	11 (9)	5 (7)	5 (17)	1 (5)
Protozoa				
*Cryptosporidium spp.*	13 (10)	10 (13)	2 (7)	1 (5)

EAEC: Enteroaggregative *E. coli*; EPEC: Enteropathogenic *E. coli*; ETEC: Enterotoxigenic *E. coli*; EHEC: Enterohemorrhagic *E. coli* 0157; STEC: Shiga toxin-expressing *E. coli*; EIEC: Enteroinvasive *E. coli*. No detection for Astrovirus, Rotavirus, *Salmonella_enterica, H. pylori, Vibrio Cholerae, Vibrio parahaemolytic, Giardia lamblia, Cryptosporidium hominus, Cryptosporidium parvum, Helminths, E. histolytica, A. Lumbricoides, N. americanus, S. Sterocoralis, T. trichiura.*

**Table 4 ijerph-16-00506-t004:** Bivariate and multivariable generalized linear mixed models of food contamination risk factors and enteric pathogen presence in infant weaning foods.

Variable	% positive (Total *n*)	Bivariate RR (95% CI)	*p*-Value	Multivariable RR (95% CI)	*p*-Value
Food					
Porridge	56 (81)	Ref	Ref	Ref	Ref
Milk	95 (20)	14.4 (1.78–116.1)	0.01	18.0 (1.85–175.6)	0.01
Non-milk/porridge	58 (26)	0.79 (0.28–2.17)	0.65	1.00 (0.33–1.12)	1
Container Type					
Covered	77 (26)	3.36 (0.57–19.9)	0.18		
Thermos	75 (24)	6.51 (1.10–38.6)	0.04		
Bottle/feeder/jug	51 (53)	2.50 (0.47–13.4)	0.28		
Uncovered	55 (11)	Ref	Ref		
Fresh	62 (13)	2.21 (0.32–15.0)	0.42		
Owning Animals					
Yes	62 (84)	1.08 (0.47–2.49)	0.85		
No	63 (43)	Ref	Ref		
Keeping Animals Inside					
Yes	59 (78)	0.74 (0.30–1.84)	0.51		
No	67 (39)	Ref	Ref		
Missing	70 (10)	None	None		
Sharing Containers					
Yes	51 (43)	0.39 (0.16–0.92)	0.03		
No	68 (84)	Ref	Ref		
Floor Permeability in Preparation Area					
Permeable	73 (26)	1.45 (0.50–4.25)	0.5		
Nonpermeable	59 (101)	Ref	Ref		
Flies in Preparation Area					
Yes	60 (40)	0.90 (0.36–2.21)	0.81		
No	62 (77)	Ref	Ref		
Feces in Preparation Area					
Yes	30 (10)	0.21 (0.04–1.00)	0.05	0.14 (0.02–0.90)	0.04
No	65 (117)	Ref	Ref	Ref	Ref
Handwash Station in Preparation Area					
Yes	69 (26)	1.58 (0.57–4.42)	0.38		
No	60 (101)	Ref	Ref		
Floor Permeability in Feeding Area					
Permeable	73 (22)	1.70 (0.55–5.25)	0.36		
Nonpermeable	60 (105)	Ref	Ref		
Flies in Feeding Area					
Yes	54 (11)	0.90 (0.41–1.98)	0.81		
No	62 (106)	Ref	Ref		
Missing	70 (10)				
Feces in Feeding Area					
Yes	60 (10)	1.23 (0.31–4.90)	0.76		
No	62 (117)	Ref	Ref		
Handwash Station in Feeding Area					
Yes	68 (19)	1.70 (0.54–5.28)	0.36		
No	61 (108)	Ref	Ref		

RR: risk ratio; CI: confidence interval; Ref: reference.

**Table 5 ijerph-16-00506-t005:** Bivariate and multivariable generalized linear mixed models of food contamination risk factors and enteric pathogen diversity in infant weaning foods.

Variable	Median (Range) Pathogen Types	Bivariate RR (95% CI)	*p*-Value	Multivariable RR (95% CI)	*p*-Value
Food					
Porridge	1 (5)	Ref	Ref	Ref	Ref
Milk	3 (9)	2.35 (1.67–3.29)	<0.001	2.35 (1.67–3.29)	<0.001
Non-milk/porridge	1 (5)	0.76 (0.50–1.12)	0.21	0.76 (0.50–1.12)	0.21
Container Type					
Covered	2.5 (9)	1.67 (0.92–3.00)	0.09		
Thermos	1 (5)	1.59 (0.82–3.07)	0.17		
Bottle/feeder/jug	1 (4)	1.41 (0.74–2.68)	0.29		
Fresh	1 (3)	0.93 (0.43–2.03)	0.86		
Uncovered	2 (5)	Ref	Ref		
Owning Animals					
Yes	1 (9)	1.29 (0.94–1.78)	0.12		
No	1 (5)	Ref	Ref		
Keeping Animals Inside					
Yes	1 (9)	1.09 (0.76–1.57)	0.62		
No	1 (5)	Ref	Ref		
Missing	Missing	Missing (Missing)			
Sharing Containers					
Yes	1 (9)	0.66 (0.46–0.96)	0.03		
No	1 (5)	Ref	Ref		
Floor Permeability in Preparation Area					
Permeable	2 (5)	0.95 (0.63–1.42)	0.8		
Non-permeable	1 (9)	Ref	Ref		
Flies in Preparation Area					
Yes	1 (5)	0.93 (0.64–1.35)	0.7		
No	1 (9)	Ref	Ref		
Missing	Missing	Missing (Missing)			
Feces in Preparation Area					
Yes	1 (4)	0.68 (0.33–1.41)	0.3		
No	1 (9)	Ref	Ref		
Handwash Station in Preparation Area					
Yes	1.5 (4)	1.29 (0.88–1.91)	0.19		
No	1 (9)	Ref	Ref		
Floor Permeability in Feeding Area					
Permeable	2 (5)	0.99 (0.64–1.51)	0.96		
Non-permeable	1 (9)	Ref	Ref		
Flies in Feeding Area					
Yes	1 (9)	1.13 (0.73–1.75)	0.58		
No	1 (5)	Ref	Ref		
Missing	Missing	Missing (Missing)			
Feces in Feeding Area					
Yes	1 (4)	1.31 (0.70–2.43)	0.39		
No	1 (9)	Ref	Ref		
Handwash Station in Feeding Area					
Yes	1 (5)	1.27 (0.80–2.00)	0.32		
No	1 (9)	Ref	Ref		

RR: risk ratio; CI: confidence interval; Ref: reference.

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
