# Peer review of "Enteric Pathogen Diversity in Infant Foods in Low-Income Neighborhoods of Kisumu, Kenya"

_ijerph, 2019, doi:10.3390/ijerph16030506_

Round 1
Reviewer 1 Report
The manuscript is solid and well-written, and the study design accomplishes the objectives. Additional samples would have provided greater strength, but it is difficult to obtain samples in such a setting. The apparent collaboration involved in the project is good and appreciated.
Line 119 - Did you monitor the ambient temperature during shipment? Documented range possible? Not a critical matter but would be positive, if possible.
Please check line 273 - add "pathogens" between . . . multiple and during. . .?
Please check line 278 - add "in" between....many and urban. . .?
Excellent discussion bringing out strengths and challenges of work like this.
Essentially no edits - it was a pleasure to read/review.
Author Response
Reviewer 1
The manuscript is solid and well-written, and the study design accomplishes the objectives. Additional samples would have provided greater strength, but it is difficult to obtain samples in such a setting. The apparent collaboration involved in the project is good and appreciated.
Line 119 - Did you monitor the ambient temperature during shipment? Documented range possible? Not a critical matter but would be positive, if possible.
Response: We transported the samples with carry-on luggage in the passenger compartment of the plane, and did not monitor the temperature during the transportation. It would be reasonable to assume samples remained at room temperature throughout the journey. Thank you for your suggestion. We will include temperature monitoring in the future.
Please check line 273 - add "pathogens" between . . . multiple and during. . .?
Response: Thank you for pointing this out. This has been revised (now line 293).
Please check line 278 - add "in" between....many and urban. . .?
Response: Thank you for pointing this out. This has been revised (now line 299).
Excellent discussion bringing out strengths and challenges of work like this.
Essentially no edits - it was a pleasure to read/review.
Thank you!

Reviewer 2 Report
The presented study contributes to filling important knowledge gaps.
Some specific comments include:
It would be helpful to elaborate the exact ingredient of some of the surveyed foods (e.g., tea and porridge) as this helps to explain the sources of pathogens.
Was the species determined for detected Aeromonas?
Cronobacter was not included in this survey but it’s a major cause of contamination in infant foods.
Complete names of EAEC and STEC need to be provided.
Table 1 is not necessary as this can be presented by text.
Table 2 Some of the percentages don’t add up to be 100% (e.g., gender)
Author Response
It would be helpful to elaborate the exact ingredient of some of the surveyed foods (e.g., tea and porridge) as this helps to explain the sources of pathogens.
Response: This is a great point. Since we did not empirically document this, we chose to address this point in the discussion (lines 311-316) about foods containing many ingredients and the potential for milk to be a source of contamination of some of the other food types: “In this study, we classified food by it’s primary ingredient, however tea is typically made with milk and sugar and porridge (made from maize meal, sorgum meal alone, or sorgum mixed with millet meal) also typically contains sugar and milk.” Also “If the latter is more common, some of the tea and porridge tested in this study may contain pathogens from milk sources.”
Was the species determined for detected Aeromonas?
Response: Line 232, 240, 288 and (current) Table 3 have been modified to specify that the primers and probes detected Aeromonas hydrophilla.
Cronobacter was not included in this survey but it’s a major cause of contamination in infant foods.
Response: Thank you for this suggestion. We did not consider Cronobacter in the design of this study, but will consider whether we can reanalyze some of our samples to quantify Cronobacter frequency.
Complete names of EAEC and STEC need to be provided.
Response: Lines 152-153 have been revised to spell out pathogenic E. coli names and the footnote for (current) Table 3 has been modified to provide full nomenclature and acronyms for pathogenic E. coli.
Table 1 is not necessary as this can be presented by text.
Response: Table 1 has been converted into text (Page 4, lines 165-176).
Table 2 Some of the percentages don’t add up to be 100% (e.g., gender)
Response: Gender was a reporting error and this has been corrected. The percentages for village are an inevitable result of rounding. These remain unadjusted to avoid reporting inaccurately and to avoid adopting decimals points that imply a higher degree of precision than reliable for 127 samples.
